# *Prune, Then Select*: SELECT HIGH-QUALITY, IMPORTANT, AND DIVERSE DATA USING TRAINING TRAJECTORIES

## ABSTRACT

The rapid expansion of instruction datasets not only escalates the computational cost of instruction fine-tuning but also brings data-related challenges, such as the presence of noisy or low-quality samples and the redundancy caused by duplicate or highly similar instances. To address these issues, data selection methods have been proposed to reduce training expenses while preserving, or even enhancing, model performance through fine-tuning on an appropriately chosen subset. In this paper, we propose a new method named **PS**, containing a **P**rune step and a **S**elect step, to ensure selecting a high-quality, important, and diverse subset by efficiently utilizing the training trajectories of data samples collected from a small proxy model. Specifically, in the **P**rune step, we prune low-quality data that do not exhibit a downward trend in their **loss trajectories**, as these samples may negatively impact the model training. In the **S**elect step, we introduce the concept of **the learning trajectory** (i.e., the loss reduction trajectory or the loss reduction rate trajectory), which provides a better representation of the model's learning progress on each data sample, and use these **learning trajectories** as sample features to cluster the retained samples from the **P**rune step. A balanced selection is then performed across all clusters within a fixed budget. We validate **PS** on the MathInstruct dataset (262K) with the open-source model suite Pythia by comparing it against two categories of data selection methods: importance-based and diversity-based methods. Experimental results show that our **PS** consistently outperforms all baseline methods across budget constraints of 30K (11.5%), 50K (19.1%), and 100K (38.2%). Notably, **PS** achieves superior performance with less than 40% of the data compared to the model trained on the full dataset.

## 1 INTRODUCTION

Fine-tuning Large Language Models (LLMs) on instruction datasets, known as instruction tuning, can unlock the potential of LLMs, enabling them to accomplish a wide variety of tasks by following natural language instructions (Ouyang et al., 2022; Taori et al., 2023). Consequently, many efforts have been devoted to collecting increasingly larger instruction-tuning datasets (Longpre et al., 2023; Yue et al., 2023; 2024; Zhang et al., 2024), either through manual collection, annotation, and transformation (Wei et al., 2022) or through automated synthesis methods (Wang et al., 2023; Taori et al., 2023), aiming to build effective instruction-following models. However, the ever-growing size of instruction datasets raises several significant challenges. First, fine-tuning LLMs on massive datasets incurs rapidly escalating computational costs. Second, both manually collected and automatically generated instruction data often contain noisy or low-quality samples (Mindermann et al., 2022). Third, when data is abundant, duplicate or highly similar samples are likely to exist, which might increase the risk of overfitting and reduce training efficiency (Lee et al., 2022). **Data selection** (Qin et al., 2024; Liu et al., 2025) is a promising direction to address these issues, which aims to carefully identify a subset from the raw dataset within a given budget such that the model trained on this subset achieves comparable or even better performance than one trained on the full dataset, while simultaneously reducing computational costs.

Existing data selection approaches can be broadly grouped into three categories: quality-based, importance-based, and diversity-based methods (Qin et al., 2024). Quality-based methods retain

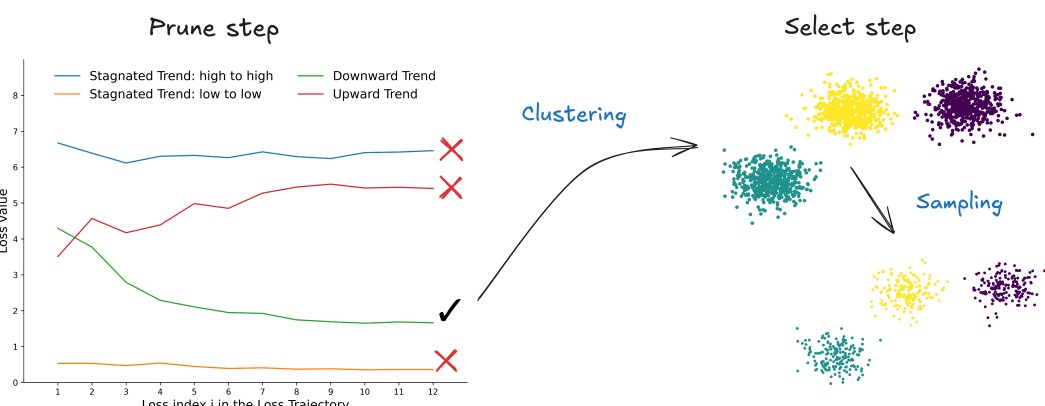

Figure 1: The overview of our proposed **PS**, which consists of a **P**rune step and a **S**elect step. In the **P**rune step, we first identify low-quality data by analyzing each data sample's **loss trajectory**. Specifically, we categorize each training sample into one of three categories: downward trend, stagnated trend, and upward trend, based on its loss trend during model training. Since low-quality samples typically show no decrease or even an increase in loss over time, they hinder training efficiency or degrade model performance. We therefore prune samples with stagnated or upward trends (see Section 4.1 for details) to guarantee the quality and importance of the selected subset. In the **S**elect step, we adopt **the learning trajectory**, a variant of the loss trajectory defined in Equation 3, as sample features to cluster the remaining samples. Then, under a fixed budget, we perform balanced sampling across all clusters to ensure diversity in the final subset (see Section 4.2 for details).

high-quality samples while discarding low-quality ones, typically by leveraging advanced LLMs to assign quality scores (Chen et al., 2024). Importance-based methods estimate sample importance using **intrinsic model metrics such as perplexity, loss, or gradients** (Jiang et al., 2019; Killamsetty et al., 2021). Diversity-based methods reduce data redundancy by clustering samples and selecting a portion from each cluster to obtain a representative subset (Sener & Savarese, 2018; Tirumala et al., 2023). Although these methods have shown promising results, focusing on a single dimension—quality, importance, or diversity—is insufficient for effective data selection. For instance, diversity-based methods promote the overall diversity of the subset but often overlook individual sample quality, potentially introducing low-quality data. Conversely, quality-based and importance-based methods emphasize sample-level quality or importance while neglecting global diversity, which can harm generalization. In addition, quality-based methods typically define "quality" through semantic attributes (e.g., grammatical correctness, fluency), while neglecting intrinsic model metrics, such as those leveraged by importance-based methods, that are critical for performance. **We therefore argue that an effective data selection strategy should integrate quality, importance, and diversity rather than relying on any single dimension.**

To identify an effective subset that is simultaneously high-quality, important, and diverse, a straightforward idea is to naively combine existing approaches. However, these different methods often rely on incompatible strategies for analyzing data samples, and combining them directly can introduce considerable computational overhead, which may negate the benefits of data selection. To address this, we propose a data selection method named **PS**, which guarantees the quality, importance, and diversity of the selected subset by efficiently leveraging data samples' loss trajectories-the sequences of loss values computed on each sample using intermediate model checkpoints during training. **PS** consists of a **P**rune step and a **S**elect step, illustrated in Figure 1. In the **P**rune step, we first analyze the loss trend of each data sample by fitting its loss trajectory with linear regression. High-quality samples typically exhibit a consistent downward trend in loss during training, indicating their importance for model learning from the perspective of intrinsic model metrics. In contrast, low-quality samples show no decrease or even an increase in loss over time, thereby hindering training efficiency or degrading model performance. We therefore prune samples without a downward trend in their **loss trajectories**, ensuring that the retained subset is both high-quality and important. To reduce additional computational cost, these loss trajectories are derived from a small proxy model rather than the target model. In the **S**elect step, we introduce the concept of the **learn-**

ing trajectory—a variant of the loss trajectory that captures the model's learning progress on each sample more directly—and use the learning trajectories as sample features to cluster the retained samples from the **P**rune step, enabling precise clustering results. Since learning trajectories can be derived directly from loss trajectories, this step introduces no additional overhead. We provide two types of learning trajectories: the loss reduction trajectory and the loss reduction rate trajectory. Finally, given a fixed budget, we perform balanced selection across clusters to ensure diversity in the final subset.

We compare our **PS** against two types of data selection methods: importance-based and diversity-based methods on the MathInstruct dataset (Yue et al., 2023) by employing the open-source model suite Pythia (Biderman et al., 2023), where Pythia-70M serves as the small proxy model for generating loss trajectories and learning trajectories for all data samples and Pythia-410M acts as the target model. The experimental results demonstrate that our **PS** consistently outperforms all baseline methods under different budget constraints of 30K (11.5%), 50K (19.1%), and 100K (38.2%). Notably, with only 100K samples (less than 40% of the full dataset), **PS** achieves better performance than training on the entire 262K dataset. The advantages of **PS** are especially pronounced under smaller budgets (e.g., 30K, 50K), highlighting that learning trajectories provide more informative features for clustering, thereby enabling more effective data selection.

## 2    RELATED WORK

**Data Selection.**    Data selection is a long-standing problem and can generally be categorized into importance-based, diversity-based, and quality-based methods (Qin et al., 2024). In importance-based approaches, each data sample will get an importance score where the importance criteria mainly rely on factors such as the perplexity (Marion et al., 2023), the error or loss of each sample (Jiang et al., 2019; Mindermann et al., 2022; Paul et al., 2023), the gradient information during model training (Killamsetty et al., 2021; Paul et al., 2023), or the influence of a sample on the predictions of other samples (Pruthi et al., 2020; Xia et al., 2024). Diversity-based data selection approaches try to select a representative subset to cover all diversities within the dataset (Sener & Savarese, 2018; Sorscher et al., 2023; Tirumala et al., 2023; Wu et al., 2023). They typically divide all data samples into distinct clusters based on their sample features like embeddings (Sorscher et al., 2023; Tirumala et al., 2023) and then select a portion of data from each cluster. Quality-based approaches involve either manually curating high-quality instruction data (Zhou et al., 2023a) or using advanced language models, such as ChatGPT or GPT-4 (OpenAI, 2023), to assign a quality score to each data example (Chen et al., 2024).

**Training Trajectories of Language Models.**    Investigating the training trajectories of language models is essential for understanding how they perform and identifying ways to improve them. Xia et al. (2023) examines how the token-level training trajectories change as language models get larger. By analyzing the intermediate training checkpoints of differently sized OPT models (Zhang et al., 2022)—from 125M to 175B parameters—on the next-token prediction task, they find that training trajectories of tokens from differently-sized models largely overlap when plotting against validation perplexity, indicating that differently-sized models make similar predictions at a similar perplexity [1]. Lin et al. (2024) also analyzes the token-level training dynamics [2] of language models for selecting useful tokens that should be aligned with the desired distribution to boost the model performance. Inspired by Xia et al. (2023), Yang et al. (2024) study the training trajectories across scales at a sample level. They empirically show that we can find groups of examples with similar training dynamics on large models by clustering the training trajectories of data samples collected from a smaller model.

## 3    PRELIMINARY

In this section, we introduce the loss trajectory and the learning trajectory of a data sample, both of which capture critical characteristics of the data sample throughout training and serve as the basis for the **P**rune and **S**elect steps of our method, respectively.

---

[1]See Appendix B.5 of Xia et al. (2023).

[2]It refers to training trajectories in these works.

### 3.1 NOTATIONS

Let $\boldsymbol{\theta} \in R^n$ denote the parameters of a language model. We consider a training instruction dataset $D_{\text{train}}$ consisting of $N$ samples (i.e., $|D_{\text{train}}| = N$), where each data sample $d_i \in D_{\text{train}}$ is a (prompt, response) pair denoted as $(x_i, y_i)$ (i.e., $d_i = (x_i, y_i)$), $i \in [N] = \{1, \cdots, N\}$. Under a fixed budget $B$, our goal is to select a subset $S \subseteq D_{\text{train}}$ such that the number of samples in $S$ satisfies $|S| = B$. Ideally, the language model trained on the selected subset $S$ should achieve performance close to, or even better than, that of the model trained on the entire dataset $D_{\text{train}}$.

### 3.2 LOSS TRAJECTORY

For a data sample $d_i = (x_i, y_i)$, $i \in [N]$, where we assume the response $y_i$ contains $M$ tokens (i.e., $y_i = (y_{i,1}, y_{i,2}, \ldots, y_{i,M})$), and given a language model $\boldsymbol{\theta}$, its loss can be computed as follows:

$$\ell_i = \ell(d_i; \boldsymbol{\theta}) = -\log p_{\boldsymbol{\theta}}(y_i \mid x_i) = -\sum_{m=1}^{M} \log p_{\boldsymbol{\theta}}(y_{i,m} \mid x_i, y_{i,1:m-1}). \tag{1}$$

This loss value depends jointly on the model parameters $\boldsymbol{\theta}$ and the data sample $d_i$. At a single training step, it provides a static measure of how well the current model fits the sample. To obtain a more informative view, we can trace how the loss evolves over successive training steps, forming **the loss trajectory** for the data sample. From the model's perspective, the loss trajectory reflects how the model gradually adapts to a specific data sample; from the data's perspective, it reveals how that sample contributes to the model's overall learning process.

Ideally, the most accurate way to obtain a loss trajectory is to record the loss of each data sample at every training step, which would precisely capture its interaction with the model throughout training. However, this is prohibitively expensive in terms of both storage and computation, so we adopt a coarse but effective approximation. Specifically, we fine-tune the language model $\boldsymbol{\theta}$ on the instruction dataset $D_{\text{train}}$ and save $T$ intermediate checkpoints $\boldsymbol{\theta}^t$, $t \in [T] = \{1, \ldots, T\}$ during training. The loss trajectory of a data sample $d_i$ is then approximated as a $T$-dimensional vector

$$\text{Loss trajectory} : \left(\ell_i^1, \ell_i^2, \ldots, \ell_i^T\right) \tag{2}$$

where each element $\ell_i^t$ is the loss computed by Equation 1 using intermediate checkpoint $\boldsymbol{\theta}^t$ on $d_i$. In our setting, each $\boldsymbol{\theta}^t$ is obtained after training for a fixed number of iterations [3] starting from $\boldsymbol{\theta}^{t-1}$ and can be saved sequentially during the training process of $\boldsymbol{\theta}$.

### 3.3 LEARNING TRAJECTORY

Although the loss trajectory of a data sample provides valuable information about how the sample contributes to the model's learning during training, it does not directly reflect the model's progress in learning that sample. For example, while a decreasing loss does suggest improvement, the magnitude of improvement between successive training steps, i.e., how much the loss is reduced, often provides a clearer signal of the model's learning progress. This motivates us to move beyond raw loss trajectories and instead construct **learning trajectories** that explicitly capture the model's learning progress on each sample.

Inspired by findings in Multi-task Learning (MTL), where loss reduction or loss reduction rate has been shown to more accurately indicate task progress than raw loss values (Désidéri, 2012; Liu et al., 2021a;b; 2023), we extend this idea to the sample level by defining two types of learning trajectories—the loss reduction trajectory and the loss reduction rate trajectory—both derived as variants of the loss trajectory:

$$\text{Learning trajectory} : (\hat{\ell}^1, \ldots, \hat{\ell}^{T-1}), \text{ where } \hat{\ell}_i = \underbrace{\ell^i - \ell^{i+1}}_{\text{loss reduction}} \text{ or } \underbrace{\frac{\ell^i - \ell^{i+1}}{\ell^i}}_{\text{loss reduction rate}} . \tag{3}$$

Equation 3 is also a coarse but effective approximation.

---

[3] Alternatively, checkpoints may be saved at different interval steps.

---

**Algorithm 1 PS**

---

**Input:** Training dataset $D_{\text{train}}$ with corresponding training trajectories $\{(\ell_i^1, \ell_i^2, \ldots, \ell_i^T)\}_{i=1}^N$, a positive threshold $h$, a fixed data budget $B$, the number of clusters $K$;
**Initialization:** an empty set $S_{\text{tmp}}$, an empty set $S$;

1: **Prune step:**
2: **for** each data sample $d_i \in D_{\text{train}}$ **do**
3:     Fit its loss trend $a_i$ with **the loss trajectory** $(\ell_i^1, \ell_i^2, \ldots, \ell_i^T)$ using Equation 4;
4:     **if** $a_i \le -h$ **then**
5:         $S_{\text{tmp}} \leftarrow S_{\text{tmp}} \bigcup d_i$.
6:     **end if**
7: **end for**
8: **Select step:**
9: Run a clustering algorithm on **the learning trajectories** $(\hat{\ell}^1, \ldots, \hat{\ell}^{T-1})$ (see Equation 3) of data examples in $S_{\text{tmp}}$ to form $K$ clusters $\mathcal{C} = \{C_1, C_2, \ldots, C_K\}$;
10: **for** each cluster $C_k$ in $\mathcal{C}$ **do**
11:     Calculate $R_k = (B - |S|)/(K - k + 1)$, the number of examples that will be sampled from $C_k$;
12:     **if** $|C_k| \le R_k$ **then**
13:         $S \leftarrow S \bigcup C_k$.
14:     **else**
15:         Sample a subset $S_k \subset C_k$ randomly, here $|S_k| = R_k$; then $S \leftarrow S \bigcup S_k$.
16:     **end if**
17: **end for**
18: Return $S$

---

# 4 METHODOLOGY

Next, we detail how our method **PS** guarantees the quality, importance, and diversity of the selected subset by efficiently leveraging the loss trajectories of data samples. **PS** consists of two steps: the **P**rune step (Section 4.1), which identifies a high-quality and important subset by exploiting loss trajectories, and the **S**elect step (Section 4.2), which refines this subset by enforcing diversity through learning trajectories, derived as variants of loss trajectories.

## 4.1 PRUNE STEP

In this step, we aim to use loss trajectories of data samples to identify a high-quality and important subset from the raw dataset. As discussed in Section 3.2, the loss trajectory of a data example reflects how it contributes to the model's learning process during training. In general, high-quality data samples facilitate learning, exhibiting a consistent downward trend in their loss as training progresses. In contrast, noisy or low-quality samples hinder the training process, reducing training efficiency or degrading model performance, typically showing no decrease or even an increase in loss over time. From the perspective of intrinsic model metrics, such samples are also unimportant, since they provide no learning signal or even misleading learning signals to model training. Therefore, by analyzing the trend of loss trajectories, we can identify these low-quality and unimportant samples and prune them, retaining only those that are both high-quality and important for effective training.

However, the raw loss trajectory of a data sample can fluctuate significantly across training steps, which complicates and undermines direct analysis. To address this issue, we approximate each trajectory with linear regression, using the slope as an efficient indicator of its overall trend. Formally, given the trajectory $\left( \ell_i^1, \ell_i^2, \ldots, \ell_i^T \right)$ for a data sample $d_i \in D_{\text{train}}$, we estimate its slope by minimizing the squared error between observed and predicted loss values:

$$f(a_i, b_i; d_i) = \min_{a,b} \sum_{j=1}^T \left( \ell_i^j - (a_i j + b_i) \right)^2. \tag{4}$$

With the slope $a_i$, we classify each training sample $d_i$ into one of three categories with respect to a predefined threshold $h > 0$:

- **Downward Trend.** For each sample $d_i$ with $a_i < -h$, we classify its loss trajectory as having a downward trend. This type of data is learnable for the model and should be retained.

- **Stagnated Tend.** If $-h \leq a_i \leq h$, the loss trajectory of $d_i$ is classified as exhibiting a stagnated trend. It can be further divided into two cases: (i) high to high: loss remains consistently high; (ii) low to low: loss remains consistently low. In both cases, these samples should be pruned as they do not contribute effectively to model learning.

- **Upward Trend.** If $h < a_i$, the sample's loss trajectory shows an upward trend. These samples should be pruned as they negatively impact the model's learning process.

This classification method is inspired by prior work (Xia et al., 2023; Lin et al., 2024), but differs in that we apply it at the sample level rather than the token level.

While this procedure provides a principled way to prune low-quality and unimportant samples, computing loss trajectories for all data with the target model is computationally expensive. To reduce this additional overhead and avoid negating the computational benefits of data selection, we leverage loss trajectories from a small model to guide our pruning step more efficiently. This strategy is supported by prior evidence: Lin et al. (2024) adopt the same approach for token-level selection, and Yang et al. (2024) show that data samples clustered by their training trajectories from a small model tend to exhibit similar training dynamics in larger models. Together, these findings suggest that small-model trajectories can serve as reliable proxies for their large-model counterparts.

### 4.2 SELECT STEP

In this step, we further refine the subset obtained from the **P**rune step. Specifically, we adopt the framework of diversity-based data selection methods to ensure that the final subset maintains sufficient diversity. Diversity-based methods typically group data samples into clusters based on their features and then select a portion from each cluster. Since their effectiveness is highly sensitive to the choice of clustering features, the representation of data samples plays a critical role. Traditional approaches often use embeddings as sample features (Sorscher et al., 2023; Tirumala et al., 2023), which primarily capture semantic attributes. Recent studies (Chen et al., 2023; Yang et al., 2024) show that clustering based on loss trajectories yields more reliable groupings, and interpret such clusters as "skills" that reflect different levels of knowledge. However, as discussed in Section 3.3, loss trajectories, while informative, do not directly reflect the model's progress in learning each sample. We argue that sample similarity should be measured by the model's progress in learning each sample, as captured by its learning trajectory.

To this end, we propose to use learning trajectories as clustering features, providing a more precise basis for forming clusters or skills. This design introduces no additional overhead, since learning trajectories can be derived directly from loss trajectories. After generating a set of clusters $\{C_1, C_2, \ldots, C_K\}$ through a clustering algorithm such as K-means, we perform a balanced sampling across clusters within a fixed budget (lines 9–17 of Algorithm 1). This ensures that each cluster contributes equally to the final subset, preventing the selection from being dominated by a few large clusters and preserving coverage across diverse learning patterns. The overall procedure of **PS** is summarized in Algorithm 1.

## 5 EXPERIMENTS

### 5.1 EXPERIMENTAL SETUP

**Baselines.** Besides (1) Random Sampling, randomly selecting samples from the given dataset, we also compare our method **PS** against two types of data selection methods: importance-based methods and diversity-based methods. **The importance-based methods** include: (2) Least Confidence (Bhatt et al., 2024) selection, which measures the model's confidence as the product of probabilities of the generated response given the prompt; (3) Middle Perplexity (Marion et al., 2023) selection, which selects samples with moderate perplexity values; (4) High Learnability, defined by the loss decrease before and after full fine-tuning (Zhou et al., 2023b); and (5) Confidence Curriculum, proposed by Varshney et al. (2022), which selects examples with decreasing confidence scores averaged over the past few epochs while mixing in a certain fraction of higher-confidence examples from previous rounds. **The diversity-based methods** include: (6) Facility Locations (Bhatt et al.,

2024), which uses the last hidden states as features; (7) DiverseEvol (Wu et al., 2023), which utilizes the embeddings. Both methods employ a $K$-Center-based strategy (Sener & Savarese, 2018) that chooses $K$ examples as centers of balls with equal radius, aiming to select a diverse subset but utilizing different sample features. Additionally, (8) SMALLTOLARGE (S2L) (Yang et al., 2024) characterizes samples via their loss trajectories. This method serves as our primary comparison baseline, as it can be viewed as the **S**elect step of our approach, with the key distinction that we exploit learning trajectories rather than loss trajectories as the sample representation.

**Training Dataset.** We follow SMALLTOLARGE (S2L) (Yang et al., 2024) to use the MathInstruct dataset (Yue et al., 2023) to validate the effectiveness of our **PS**. It is meticulously compiled from 14 math datasets [4], comprising a total of 262K training examples, ensuring extensive coverage across diverse fields of math. Moreover, the dataset integrates chain-of-thought (CoT) and program-of-thought (PoT) rationales, facilitating the effective use of tools and enabling diverse thought processes tailored to various math problems.

**Evaluation benchmarks.** We also follow Yang et al. (2024) to evaluate all methods on three in-domain datasets, including GSM8K (Cobbe et al., 2021), MATH (Hendrycks et al., 2021), and NumGLUE (Mishra et al., 2022) and three out-of-domain datasets, containing SVAMP (Patel et al., 2021), Mathematics (Davies et al., 2021), SimulEq (Koncel-Kedziorski et al., 2016). These chosen evaluation datasets consist of open-formed questions and cover diverse mathematical subjects, including calculus, algebra, probability, number theory, and geometry.

**Evaluation metric.** We adopt the standard evaluation metric: exact match for open-formed evaluation benchmarks. This metric evaluates the model's accuracy by determining whether its generated answers precisely match the correct solutions. An answer is considered correct only if it exactly matches the reference solution [5].

## 5.2 IMPLEMENTATION DETAILS

SMALLTOLARGE (S2L) (Yang et al., 2024) is our primary comparison baseline. They have implemented the baseline methods from (1) to (7) and demonstrate that S2L outperforms all of them. For convenience, we reproduce only the experimental results of S2L by strictly following the implementation details reported in the original paper and implement our proposed **PS**.

**Training details.** We employ the open-source model suite Pythia (Biderman et al., 2023) as our base model for conducting the experiments. Pythia-70M serves as the small proxy model used to generate the loss trajectories and learning trajectories for all training data samples where we save all middle checkpoints every 500 steps, which is suitable for both S2L and our **PS**. Pythia-410M acts as the target model for comparing different data selection methods. To ensure a fair comparison, we maintain a consistent training schedule across all methods, varying model sizes, and different data budgets. Specifically, all models are trained with a batch size of 128 and a maximum sequence length of 512. The number of training steps of all experiments is standardized to correspond to 3 epochs on the respective dataset, including the full dataset and different selected subsets, with a learning rate of 2e-5, and a cosine learning rate scheduler with a 3% warm-up period.

**Selection details.** In the **P**rune step, we employ the linear regression algorithm from the scikit-learn package to fit the loss trends of all data samples using the default settings. A threshold of $h = 0.02$ is applied to filter out low-quality data samples (i.e., the data point $d_i$ with a slope $a_i \geq -0.02$ will be discarded). Finally, we prune 31K low-quality data samples (i.e., 12% of the full dataset), most of which exhibit a stagnated trend, and keep 230K high-quality samples. In the **S**elect step, we use the K-means algorithm provided by the faiss package [6] with $K = 100$ to cluster each source separately for 14 different sources of the MathInstruct dataset (Yue et al., 2023), same for reproducing S2L. In Yang et al. (2024), it was observed that this per-source selection benefits S2L, as different data sources within MathInstruct display distinct common patterns in their training trajectories. We compare all methods under three budget constraints: $B = 30K$, 50K, and 100K.

---

[4] https://huggingface.co/datasets/TIGER-Lab/MathInstruct.

[5] https://github.com/TIGER-AI-Lab/MAmmoTH.

[6] https://github.com/facebookresearch/faiss/tree/main.

Table 1: The performance of all data selection methods evaluated on both in-domain and out-of-domain datasets given three budget constraints of 30K, 50K, and 100K. Pythia-410M serves as the base model. To ensure a fair evaluation between S2L and **PS**, we sample three subsets for each method and each budget and train the model on each subset with three different seeds. Consequently, the results are averaged over nine models. The results for NONE are also averaged from three runs with different seeds. The highest average accuracies in each budget are bold. We use the loss reduction trajectory as the learning trajectory to report the results of our **PS**. We provide the results of using the loss reduction rate trajectory as the learning trajectory of **PS** in Table 3.

| Methods | | Budget | In-domain | | | | Out-of-domain | | | |
|---|---|---|---|---|---|---|---|---|---|---|
| | | | GSM8K | MATH | NumGLUE | **Avg.** | SVAMP | Mathematics | SimulEq | **Avg.** |
| | (PRETRAINED) | | 2.0 | 1.6 | 10.1 | 4.6 | 2.3 | 2.5 | 1.4 | 3.3 |
| | RANDOM | 30K | 3.3 | 6.2 | 15.0 | 8.2 | 15.0 | 15.1 | 1.6 | 9.4 |
| | | 50K | 3.7 | 6.4 | 18.1 | 9.4 | 17.0 | 11.6 | 1.2 | 9.7 |
| | | 100K | 5.9 | 7.6 | 22.0 | 11.8 | 20.5 | 20.8 | 2.7 | 13.3 |
| **Importance-based** Methods | LEAST CONFIDENCE | 30K | 2.7 | 1.3 | 18.0 | 7.0 | 13.7 | 3.3 | 1.4 | 6.7 |
| | | 50K | 2.1 | 1.7 | 21.0 | 8.3 | 14.5 | 3.5 | 1.0 | 7.3 |
| | | 100K | 2.5 | 3.3 | 23.5 | 9.8 | 20.8 | 6.3 | 3.7 | 10.0 |
| | MIDDLE PERPLEXITY | 30K | 5.3 | 3.7 | 16.2 | 8.4 | 14.2 | 8.7 | 1.2 | 8.2 |
| | | 50K | 3.2 | 5.9 | 20.5 | 9.9 | 18.1 | 11.3 | 5.1 | 10.7 |
| | | 100K | 5.4 | 7.2 | 20.9 | 11.2 | 23.8 | 15.3 | 3.3 | 12.6 |
| | HIGH LEARNABILITY | 30K | 6.1 | 1.6 | 19.1 | 8.9 | 10.7 | 9.9 | 1.4 | 8.1 |
| | | 50K | 6.1 | 2.1 | 18.6 | 8.9 | 14.5 | 14.0 | 2.1 | 8.9 |
| | | 100K | 7.4 | 9.2 | 29.8 | 15.5 | 20.7 | 19.4 | 10.3 | 16.1 |
| | CONFIDENCE CURRICULUM | 30K | 4.2 | 6.3 | 15.4 | 8.6 | 18.9 | 16.5 | 1.4 | 10.4 |
| | | 50K | 6.6 | 3.3 | 16.9 | 9.0 | 19.9 | 19.6 | 2.1 | 11.4 |
| | | 100K | 4.6 | 6.3 | 17.1 | 9.3 | 21.0 | 15.2 | 1.8 | 11.0 |
| **Diversity-based** Methods | FACILITY LOCATIONS | 30K | 4.2 | 7.7 | 10.0 | 7.3 | 11.8 | 13.8 | 1.2 | 8.1 |
| | | 50K | 5.7 | 9.1 | 12.4 | 9.1 | 15.4 | 18.6 | 1.6 | 10.5 |
| | | 100K | 7.4 | 10.9 | 30.5 | 16.3 | 26.2 | 21.9 | 9.3 | 17.7 |
| | DIVERSEEVOL | 30K | 1.9 | 3.6 | 8.8 | 4.8 | 13.9 | 3.0 | 1.6 | 5.5 |
| | | 50K | 1.6 | 4.2 | 12.0 | 5.9 | 10.6 | 7.3 | 1.9 | 6.3 |
| | | 100K | 1.3 | 3.8 | 12.9 | 6.0 | 11.8 | 8.4 | 1.0 | 6.5 |
| | S2L[†] | 30K | 3.5 | 7.0 | 16.3 | 9.0 | 17.5 | 17.7 | 1.3 | 12.1 |
| | | 50K | 5.4 | 8.7 | 22.0 | 12.0 | 21.5 | 19.2 | 4.8 | 15.2 |
| | | 100K | 8.8 | 11.3 | 29.9 | **16.7** | 26.1 | 23.0 | 9.6 | 19.6 |
| **Combined** | **PS**[†] (Ours) | 30K | 3.4 | 7.5 | 19.2 | **10.0** | 19.9 | 17.6 | 2.1 | **13.2** |
| | | 50K | 5.6 | 9.2 | 23.3 | **12.7** | 23.9 | 19.4 | 4.7 | **16.0** |
| | | 100K | 9.2 | 11.2 | 29.3 | 16.6 | 28.0 | 22.9 | 9.7 | **20.2** |
| | NONE[†] | 262K | 9.0 | 10.4 | 28.8 | 16.1 | 26.7 | 24.6 | 7.0 | 19.4 |

[1] Methods with [†] are reproduced or implemented by ourselves.
[2] The results of other methods without [†] are reported by Yang et al. (2024).
[3] NONE indicates the results from the model trained on the full MathInstruct dataset.

## 5.3 RESULTS AND ANALYSIS

Table 1 shows that our **PS** consistently outperforms all baseline methods, including S2L, across the three budget constraints of 30K, 50K, and 100K. Furthermore, both S2L and our **PS** achieve better performance using less than 40% of the data (i.e., 100K samples) compared to the model trained on the full dataset (i.e., 262K samples). Note that the reported results for **PS** are based on using the loss reduction trajectory as the learning trajectory. See Table 3 for the results obtained with the loss reduction rate trajectory as the learning trajectory.

In particular, when the budget is smaller, the effectiveness of our method becomes more notable. With a 100K budget, the average accuracy of our **PS** is slightly lower than S2L on the three in-domain datasets (16.6 vs. 16.7) but surpasses S2L on the three out-of-domain datasets with an absolute improvement of 0.6 (20.2 vs. 19.6). At a 50K budget, **PS** outperforms S2L on the in-domain datasets with an absolute improvement of 0.7 (12.7 vs. 12.0) and on the out-of-domain datasets with an improvement of 0.8 (16.0 vs. 15.2). When the budget is reduced to 30K, **PS** achieves an even greater advantage, outperforming S2L by 1.0 on the in-domain datasets (10.0 vs. 9.0) and by 1.1 on the out-of-domain datasets (13.2 vs. 12.1). This indicates that using the learning trajectories as sample features to cluster the retained data samples from the **P**rune step indeed can obtain more precise clustering results. In the **S**elect step, all our selections across different budgets are carried

out on the fixed clustering results of $\{C_1, C_2, \ldots, C_K\}$ obtained by running a clustering algorithm on the retained data samples. Given a small budget, the more accurate the clustering results, the more diverse the extracted subset, ensuring better performance of the trained model on this subset.

## 5.4 ABLATION STUDY

**Loss trajectories can be used to help distinguish between high-quality and low-quality data samples**. In the Prune step, we prune 31K low-quality data samples (i.e., 12% of the full dataset) and keep 230K high-quality samples. We conduct experiments, the results of which are presented in Table 2, to validate that the data samples whose loss trajectories show a non-downward trend (i.e., stagnated trend or upward trend) are low-quality samples compared with those with downward trend loss trajectories from two perspectives.

Firstly, removing 31K data samples whose loss trajectories show a non-downward trend does not hurt the model's performance. As most of these samples exhibit a stagnated trend during training (with only 79 out of 31K samples showing an upward trend in their loss trajectories), the model's performance on the retained samples shows only a marginal improvement, as shown in the upper section of Table 2. Secondly, we compare the performance of models trained on the 30K data samples exhibiting **non-downward trend** loss trajectories with those trained on 30K samples showing **downward trend** loss trajectories. Specifically, we propose two strategies for selecting data examples with **downward trend** loss trajectories. The first is our **PS**, and the second involves selecting samples with the steepest downward

Table 2: The ablation study results for our Prune step. Steepest means selecting samples whose loss trajectories with the steepest downward trends (i.e., the smallest slope values $a_i$).

| Methods | | Budget | In-domain Avg. | Out-of-domain Avg. |
|---|---|---|---|---|
| NONE | | 262K | 16.1 | 19.4 |
| **PS**-P | | 230K | 16.2 | 19.7 |
| **Non-downward trend** | | 30K | 7.65 | 8.3 |
| **Downward trend** | Steepest | 30K | **11.1** | 11.4 |
| | **PS** | 30K | 10.0 | **13.2** |
| | Steepest | 50K | 12.0 | 13.7 |
| | **PS** | 50K | **12.7** | **16.0** |
| | Steepest | 100K | 13.3 | 16.6 |
| | **PS** | 100K | **16.6** | **20.2** |

trends (i.e., the smallest slope values $a_i$). As seen in the lower section of Table 2, the models trained on the 30K data samples exhibiting **non-downward trend** loss trajectories have worse performance, confirming that these samples are of lower quality compared to those with **downward trend** loss trajectories. Moreover, **PS** outperforms the strategy of selecting data examples with the steepest downward trends, highlighting the importance of selecting a diverse subset of data samples.

## 6 CONCLUSION

In this paper, we propose a data selection method named **PS**, which consists of a Prune step and a Select step, to obtain a high-quality, important, and diverse subset by leveraging the training trajectories of data samples collected from a small proxy model. In the Prune step, we analyze each sample's loss trajectory to identify and prune low-quality data. In the Select step, we introduce the learning trajectory as a more informative sample feature for clustering and then perform balanced sampling across all clusters with a fixed budget. We validate **PS** on the MathInstruct dataset with the open-source model suite Pythia by comparing it against eight data selection methods. Our **PS** consistently outperforms all baselines across budget constraints of 30K, 50K, and 100K, with particularly strong gains under smaller budgets (e.g., 30K, 50K), demonstrating the effectiveness of learning trajectories for diversity-based methods. Notably, with only 100K samples (less than 40% of the full dataset), **PS** achieves better performance than training on the entire 262K dataset.

## LIMITATION

We have currently validated our method only on the MathInstruct dataset using the open-source model suite Pythia. In the future, we plan to evaluate the proposed method on a broader range of datasets, especially synthetic datasets that are likely to contain a substantial amount of noisy data samples. We anticipate that our Prune step will demonstrate even greater advantages on such

datasets. Meanwhile, we will also conduct experiments with a broader range of model architectures and larger model sizes, such as the LLaMA family.

## ETHICS STATEMENT

This paper does not involve any ethics-related issues. The data and resources utilized in this work are open-source and widely adopted in numerous existing studies.

## THE USE OF LARGE LANGUAGE MODELS (LLMS)

We used large language models (LLMs) exclusively for writing assistance in the preparation of this paper, specifically to improve clarity, grammar, and readability. No research ideas, methods, analyses, or results were generated by LLMs. All scientific content is entirely the work of the authors.

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

| Methods | Budget | In-domain | | | | Out-of-domain | | | |
|---|---|---|---|---|---|---|---|---|---|
| | | GSM8K | MATH | NumGLUE | $\text{Avg}_{\text{std}}$ | SVAMP | Mathematics | SimulEq | $\text{Avg}_{\text{std}}$ |
| S2L | 30K | $3.5_{\pm0.6}$ | $7.0_{\pm0.8}$ | $16.3_{\pm1.4}$ | $9.0_{\pm0.8}$ | $17.5_{\pm1.5}$ | $17.7_{\pm2.1}$ | $1.3_{\pm0.6}$ | $12.1_{\pm0.9}$ |
| | 50K | $5.4_{\pm1.1}$ | $8.7_{\pm0.4}$ | $22.0_{\pm3.2}$ | $12.0_{\pm1.4}$ | $21.5_{\pm1.6}$ | $19.2_{\pm2.2}$ | $4.8_{\pm1.8}$ | $15.2_{\pm1.4}$ |
| | 100K | $8.8_{\pm0.7}$ | $11.3_{\pm0.4}$ | $29.9_{\pm0.9}$ | $16.7_{\pm0.5}$ | $26.1_{\pm1.3}$ | $23.0_{\pm2.3}$ | $9.6_{\pm1.3}$ | $19.6_{\pm1.1}$ |
| **PS** reduction | 30K | $3.4_{\pm0.6}$ | $7.5_{\pm0.5}$ | $19.2_{\pm1.8}$ | $10.0_{\pm0.9}$ | $19.9_{\pm1.6}$ | $17.6_{\pm1.6}$ | $2.1_{\pm0.9}$ | $13.2_{\pm0.9}$ |
| | 50K | $5.6_{\pm0.5}$ | $9.2_{\pm0.4}$ | $23.3_{\pm1.6}$ | $12.7_{\pm0.7}$ | $23.9_{\pm1.3}$ | $19.4_{\pm1.1}$ | $4.7_{\pm1.0}$ | $16.0_{\pm0.6}$ |
| | 100K | $9.2_{\pm0.7}$ | $11.2_{\pm0.3}$ | $29.3_{\pm2.3}$ | $16.6_{\pm0.7}$ | $28.0_{\pm1.3}$ | $22.9_{\pm2.6}$ | $9.7_{\pm1.3}$ | $20.2_{\pm1.2}$ |
| **PS** reduction rate | 30K | $3.6_{\pm0.5}$ | $7.9_{\pm0.3}$ | $17.2_{\pm2.4}$ | $9.6_{\pm1.0}$ | $18.4_{\pm1.3}$ | $17.6_{\pm1.5}$ | $1.6_{\pm0.4}$ | $12.5_{\pm0.7}$ |
| | 50K | $5.6_{\pm0.8}$ | $9.4_{\pm0.6}$ | $22.7_{\pm1.7}$ | $12.5_{\pm0.9}$ | $22.4_{\pm1.5}$ | $19.7_{\pm1.5}$ | $2.6_{\pm0.8}$ | $14.9_{\pm0.4}$ |
| | 100K | $9.3_{\pm0.8}$ | $11.0_{\pm0.4}$ | $29.6_{\pm1.8}$ | $16.6_{\pm0.5}$ | $27.4_{\pm1.8}$ | $22.8_{\pm2.3}$ | $7.7_{\pm2.1}$ | $19.3_{\pm1.1}$ |
| NONE | 262K | $9.0_{\pm0.6}$ | $10.4_{\pm1.0}$ | $28.8_{\pm1.9}$ | $16.1_{\pm0.8}$ | $26.7_{\pm1.3}$ | $24.6_{\pm2.4}$ | $7.0_{\pm1.6}$ | $19.4_{\pm1.0}$ |

Table 3: The performance of S2L and **PS** evaluated on both in-domain and out-of-domain datasets given three budget constraints of 30K, 50K, and 100K. Pythia-410M serves as the base model. To ensure a fair evaluation between S2L and **PS**, we sample three subsets for each method and each budget and train the model on each subset with three different seeds. Consequently, the results are averaged over nine models. The results for NONE are also averaged from three runs with different seeds.

# A  MORE RESULTS

Table 3 is the results of **PS** reduction rate obtained with the loss reduction rate trajectory as the learning trajectory. At a 30K budget, **PS** reduction rate outperforms S2L on the in-domain datasets with an absolute improvement of 0.6 (9.6 vs. 9.0) and on the out-of-domain datasets with an improvement of 0.4 (12.5 vs. 12.1). At a 50K budget, **PS** reduction rate surpasses S2L on the in-domain datasets with an absolute improvement of 0.5 (12.5 vs. 12.0) while slightly trailing S2L on the out-of-domain datasets (14.9 vs. 15.2). With a 100K budget, the average accuracy of **PS** reduction rate is slightly lower than S2L.

