# OpenReview forum: "Prune, Then Select: Select High-Quality, Important, and Diverse Data Using Training Trajectories"
_ICLR.cc/2026/Conference — ICLR 2026 Conference Withdrawn Submission_

### Official Review · Reviewer_WNFj · 2025-10-18

**Soundness:** 2
**Presentation:** 2
**Contribution:** 3
**Rating:** 4
**Confidence:** 4

**Summary:**

This paper proposes PS (Prune, then Select), a data selection method for instruction fine-tuning that combines quality filtering and diversity-based selection. The method operates in two steps: (1) a Prune step that analyzes loss trajectories from a small proxy model to remove low-quality samples that don't exhibit downward loss trends, and (2) a Select step that clusters retained samples using "learning trajectories" (loss reduction or loss reduction rate over time) and performs balanced sampling across clusters. The authors evaluate PS on the MathInstruct dataset (262K samples) using Pythia models (70M as proxy, 410M as target), showing improvements over baseline methods including S2L at budgets of 30K, 50K, and 100K samples.

**Strengths:**

Clear motivation: The paper articulates well why combining quality, importance, and diversity is important for data selection, addressing limitations of single-dimension approaches.

Computational efficiency: Using a small proxy model (70M) to generate trajectories rather than the target model is a practical design choice that reduces computational overhead.

Consistent improvements: PS shows improvements over baselines across multiple budget constraints, with particularly strong gains at smaller budgets (30K, 50K).

Novel use of learning trajectories: The idea of using loss reduction/reduction rate trajectories instead of raw loss trajectories for clustering is interesting and inspired by multi-task learning literature.

Well-structured presentation: The paper is generally well-organized with clear algorithm description and good use of figures to illustrate the method.

**Weaknesses:**

1. Limited Experimental Validation

Single dataset: Only MathInstruct is evaluated. This is a significant limitation that the authors acknowledge but don't address. Mathematics is a specific domain with particular characteristics (e.g., verifiable answers, structured reasoning). Generalization to other domains (creative writing, coding, general instruction-following) is unclear.

Small model scale: Only tested on Pythia 70M→410M. The assumption that small model trajectories transfer to large models needs validation at realistic scales (e.g., 7B, 13B, 70B models commonly used today). The 70M to 410M jump is relatively modest.

No comparison on other datasets: Prior work (S2L, baselines) has been evaluated on other datasets. Why not compare on those?

2. Methodological Concerns

a. Hyperparameter selection and sensitivity:

The threshold h=0.02 appears arbitrary with no justification, ablation study, or sensitivity analysis. How was this chosen? How sensitive are results to this value?

K=100 clusters - no justification provided. How does performance vary with K?

Per-source clustering for 14 sources is very specific to MathInstruct structure. How does this generalize to datasets without clear source boundaries?

b. Linear regression for trend analysis:

Fitting loss trajectories with linear regression (Equation 4) may be overly simplistic. Loss curves often exhibit non-linear behavior (e.g., initial steep drop, then plateau).

No comparison with alternative trend detection methods (e.g., polynomial fitting, monotonicity tests, non-parametric trend tests).

c. Learning trajectory definition:

Equation 3 uses the same notation ℓ̂ for both loss reduction and loss reduction rate, which is confusing.

For loss reduction rate, when ℓ^i is very small, the denominator could cause numerical instability. How is this handled?

The paper doesn't provide clear guidance on when to use loss reduction vs. loss reduction rate. Tables 1 and 3 show similar performance - is there a principled selection criterion?

d. Balanced sampling assumption:

Lines 10-17 of Algorithm 1 perform balanced sampling across clusters. The paper doesn't justify why equal representation from each cluster is optimal. Some clusters might contain more important or higher-quality data.

The calculation Rk = (B - |S|)/(K - k + 1) assumes remaining budget should be distributed equally across remaining clusters, but this may not be optimal.

3. Experimental Design Issues

a. Unfair comparisons:

Most baseline results (methods 1-7) are taken directly from Yang et al. (2024) rather than reproduced. This makes it difficult to ensure fair comparison with identical experimental settings.

Only S2L and PS are reproduced. Differences in implementation details, random seeds, or training procedures could affect results.

b. Statistical significance:

Standard deviations are provided but no statistical tests (e.g., t-tests, Mann-Whitney U tests).

Many improvements are marginal: at 100K budget, in-domain average is 16.6 vs 16.7 (PS vs S2L). Is this difference statistically significant?

With 9 runs (3 subsets × 3 seeds), proper statistical testing is feasible and should be included.

c. Limited evaluation metrics:

Only exact match accuracy is reported.

No analysis of other important aspects: computational cost comparison, model calibration, robustness to distribution shift, quality of generated reasoning chains.

4. Limited Analysis and Ablations

a. Insufficient ablation studies:

Table 2 only ablates the pruning decision (downward vs non-downward trends).

Missing ablations on:

Threshold h values

Number of clusters K

Different clustering algorithms (K-means vs alternatives)

Loss reduction vs loss reduction rate trajectories

Impact of proxy model size

Per-source vs global clustering



b. Lack of qualitative analysis:

No examples of what types of samples get pruned vs retained.

No characterization of what the learned clusters represent. Do they correspond to difficulty levels, problem types, reasoning patterns?

No visualization of the learning trajectories to build intuition.

c. Missing failure analysis:

At 100K budget, PS slightly underperforms S2L on in-domain tasks (16.6 vs 16.7). Why?

What types of data benefit most/least from this selection approach?

5. Novelty Concerns

The Prune step is heavily inspired by prior token-level work (Xia et al., 2023; Lin et al., 2024), adapted to sample-level with minimal modification.

The Select step is essentially S2L with learning trajectories instead of loss trajectories - an incremental change.

The main contribution is combining these two steps, but the paper doesn't provide strong theoretical or empirical justification for why this combination is superior to alternatives (e.g., why not use learning trajectories in the Prune step too?).

6. Clarity Issues

a. Important details missing:

Total number of checkpoints T is not clearly specified. The paper says "every 500 steps" but what is T?

Computational cost comparison with baselines is missing. How much more expensive is PS compared to random sampling or S2L?

What is the total training time for the proxy model?

b. Inconsistent terminology:

"Learning trajectory" is used generically when there are two distinct variants (loss reduction vs loss reduction rate).

The benefit of learning trajectories over loss trajectories is asserted but not demonstrated through controlled comparison (e.g., using loss trajectories in Select step as a direct ablation).

**Questions:**

In Table 2, only 79 out of 31K pruned samples show upward trend. Why is this number so low? Does this suggest the upward trend criterion is too strict?

How does the method handle samples with non-monotonic loss trajectories (e.g., initial decrease, then increase, then decrease again)?

For per-source clustering with 14 sources and K=100, are there 100 clusters per source (1400 total) or 100 total? The paper is unclear.
What is the overlap between samples selected by PS at different budgets? Are they nested subsets?

How does performance scale with the proxy-to-target model size ratio? Would a 410M→7B setup still work?

---

### Official Review · Reviewer_1DGp · 2025-10-26

**Soundness:** 3
**Presentation:** 3
**Contribution:** 2
**Rating:** 4
**Confidence:** 3

**Summary:**

This paper introduces Prune, Then Select (PS), a two-stage data selection framework for instruction fine-tuning of large language models. In the Prune stage, examples with unlearnable or noisy behaviors are removed by fitting a linear model to their loss trajectories from a small proxy model and discarding those with low or flat improvement slopes. In the Select stage, the remaining samples are clustered in training trajectory space reduction patterns and a balanced subset is drawn across clusters to encourage diversity under a fixed budget. The approach is evaluated using the Pythia model family on the MathInstruct dataset, showing that PS achieves comparable or superior performance to training on the full dataset while using less than 40% of the data.

**Strengths:**

- the paper is very clearly written that the reviewer appreciates!
- i like the ablation studies comparing the slices of sft data grouped by loss trajectory. The difference in performance is pretty obvious to support the argument that loss trajectory is useful to distinguish good/bad data subsets.

**Weaknesses:**

- I think a key issue with the paper is that it's actually not that stronger than S2L baseline for example - taking into account fact that the author will spend majority of time tuning their method and little time tuning baselines. I wonder how significant of a contribution it is from prior works. Also, the author's approach is essentially S2L baseline with prune step. Some what question the significance of the contribution put forward in this paper. The idea of using loss trajectory is not novel.
- The paper will benefit from results on different base llm, e.g., in addition to pythia and additional ablations on hyperparameters: predefined threshold, number of clusters,
- The paper currently does not support the argument that clustering in training trajectory space is superiour to semantic embedding space (using embedding model as basis to do clustering). This is important - please consider adding experiment (try a few different embedding model and make sure these baselines are tuned) showing this is actually true empirically.
- The paper would benefit a lot from qualitative examples of the groups that are pruned / kept according to their loss trajectories and overall interpretation of what are characteristics of examples that fall into each category. It would help the reader build more intuition on what the algorithm is actually doing.

**Questions:**

- Is linear function a reasonable parametric form for modeling loss trajectories ? The loss curve does not look linear to me.
- Not necessary to include in the rebuttal or run experiments, but i'd be interested to know if the proxy model is trained on a different sft dataset (similar or very different from ones used for data selection), what would be its affect on its utility as the proxy. is it important for the proxy to be finetuned on the same sft dataset? additional would a smaller model from another model family (e.g., llama) when selecting data for one model family (e.g., pythia). It would be interesting to know what we are lossing by using a smaller proxy model as opposed to the model to be finetuned

---

### Official Review · Reviewer_ykME · 2025-11-02

**Soundness:** 3
**Presentation:** 3
**Contribution:** 2
**Rating:** 2
**Confidence:** 3

**Summary:**

The paper proposes PS (Prune, then Select) for instruction-tuning data selection using training trajectories from a small proxy model.

Prune: fit a linear trend to each sample’s loss trajectory; drop samples without a clear downward slope.

Select: compute a “learning trajectory” (loss reduction or reduction rate between checkpoints), cluster retained samples, and balanced sample across clusters under a budget.
On MathInstruct (262K) with Pythia-70M→410M, PS beats importance- and diversity-based baselines (incl. S2L) under budgets 30K/50K/100K, and with 100K (<40%) matches/exceeds full-data training.

**Strengths:**

Clear, simple pipeline: two steps (quality+importance via pruning, diversity via cluster sampling) are easy to reproduce.

Proxy-trajectory idea is practical: leverages small-model checkpoints to guide selection with low cost.

Learning-trajectory feature: using delta loss rather than raw loss is a sensible, informative signal for clustering.

Consistent gains at small budgets: strongest improvements at 30K/50K—useful when compute is tight.

**Weaknesses:**

Narrow scope: only math data (MathInstruct) and Pythia 70M/410M; unclear generality to other domains (general instructions, code, safety) or larger/base models (Llama, Qwen, Mixtral).

Heuristic pruning risk: linear slope over few checkpoints can be noisy; may drop hard-but-useful samples (curriculum/long-horizon) or keep spurious “downward” ones.

Limited design ablations: sensitivity to h, T (checkpoints), K (clusters), per-source vs global clustering, and the balanced sampling rule is underexplored.

**Questions:**

There are small models such as qwen, granite that is more relevent for small models, the author should replace the baseline model pythia to modern base model to at least demonstrate the usefulness of the methods.

---

### Official Review · Reviewer_mvQ4 · 2025-11-02

**Soundness:** 2
**Presentation:** 2
**Contribution:** 2
**Rating:** 2
**Confidence:** 4

**Summary:**

This paper proposes a two-stage data selection pipeline for instruction-tuning: (i) Prune low-quality samples using the slope of each example’s loss trajectory (linear fit over intermediate checkpoints); (ii) Select a diverse subset by clustering learning trajectories (loss reductions between checkpoints) and balanced sampling per cluster. The proposed method is evaluated on several LLM fine-tuning benchmarks.

**Strengths:**

- The proposed method is simple and easy to implement
- The proposed method demonstrates consistent (although small) empirical performance gains across multiple experiment settings.

**Weaknesses:**

- The proposed method is incremental compared to S2L. The proposed method adds a pruning step by the loss trajectory slope, and the selection step is essentially the same as the S2L.
- The proposed pruning method is a heuristic that needs better justification. It also introduces additional hyperparameters (e.g., the threshold h) that should be more thoroughly analyzed.
- The performance gain over S2L is mostly small.
- The experiments do not report the computational cost comparison.
- There is a lack of apple-to-apple comparison with non-S2L baselines. Only S2L is reproduced; other baselines are taken from prior work.

**Questions:**

See Weaknesses.

---

### Note · Authors · 2025-11-23

**Comment:**

We plan to make substantial updates. Thank you to all the reviewers.

**Withdrawal Confirmation:**

I have read and agree with the venue's withdrawal policy on behalf of myself and my co-authors.